# Use of Hyaluronic Acid (HA) in Chronic Airway Diseases

**DOI:** 10.3390/cells9102210

**Published:** 2020-09-29

**Authors:** Luis Máiz Carro, Miguel A. Martínez-García

**Affiliations:** 1Chronic Bronchial Infection, Cystic Fibrosis and Bronchiectasis Unit, Ramón y Cajal University Hospital, 28001 Madrid, Spain; 2Respiratory Department, La Fe University and Polytechnic Hospital, 46001 Valencia, Spain; mianmartinezgarcia@gmail.com

**Keywords:** hyaluronic acid, hypertonic saline, emphysema, asthma, bronchiectasis, elastase, COPD, cystic fibrosis

## Abstract

Hyaluronic acid (HA) is a key component of the extracellular matrix of the lungs. A unique attribute of HA is its water-retaining properties, so HA has a major role in the regulation of fluid balance in the lung interstitium. Hyaluronic acid has been widely used in the treatment of eyes, ears, joints and skin disorders, but in the last years, it has been also proposed in the treatment of certain lung diseases, including airway diseases, due to its anti-inflammatory and water-binding capacities. Hyaluronic acid aerosol decreases the severity of elastase-induced emphysema in murine models, prevents bronchoconstriction in asthmatics and improves some functional parameters in chronic obstructive pulmonary disease (COPD) patients. Due to the protection of HA against bronchoconstriction and its hydration properties, inhaled HA would increase the volume of airway surface liquid, resulting in mucus hydration, increased mucous transport and less mucous plugging of the airways. In addition, it has been seen in human studies that the treatment with nebulised HA improves the tolerability of nebulised hypertonic saline (even at 6% or 7% of concentration), which has been demonstrated to be an effective treatment in bronchial secretion management in patients with cystic fibrosis and bronchiectasis. Our objective is to review the role of HA treatment in the management of chronic airway diseases.

## 1. Introduction

The extracellular matrix of lung tissue is formed by a combination of molecules as collagen, elastin and glycosaminoglycans, key to maintaining its stability [1,2]. Hyaluronic acid (HA), a glycosaminoglycan, is present in high concentrations in the connective tissue of the lungs [3,4] HA is synthetised by the hyaluronate synthetase and degraded by the hyaluronidases.

The properties of the HA molecule are related with its size, so there is a clear difference between HA molecules with high molecular weights (>1 million Daltons) and the small fragments with low molecular weights (150,000–300,000 Daltons) produced during inflammation. High-molecular weight HA fragments have antiangiogenic and anti-inflammatory properties, while those of low molecular weights produced during inflammation are proinflammatory and proangiogenic molecules and promote cellular migration [5,6]. Additionally, HA has the property to retain a significant quantity of water in the extracellular matrix, producing viscous gels that might play an important role both in the homeostasis of the tissues and their biomechanical integrity [7,8,9].

Some studies highlight the key role that HA and its degradation products play in the physiopathology of the respiratory system. Lung damage produces the liberation of short-fragment HA. This release activates innate immune receptors, which can result in inflammation, remodelling and hyperresponsiveness, in addition to other clinical symptoms. The defective clearance in an injury results in the development of bacterial biofilms. The perpetuation of this vicious cycle can be altered through the modulation of short-fragment HA signalling, for instance, with high-molecular weight HA (Figure 1). HA mitigates the action of elastases such as porcine pancreatic elastase, as well as human neutrophil elastase and human macrophage metalloelastase. This has been demonstrated in animal models of pulmonary emphysema, as well as in elastin matrices produced by pleural mesothelial cell cultures. Studies have indicated that small fragments of HA contribute to the immune cell response, binding to specific cell-surface receptors. Low-molecular weight fragments stimulate mouse alveolar macrophages to produce several cytokines, including metalloelastase. High-molecular weight fragments suppressed such expressions.

Furthermore, high-molecular weight HA administered via aerosol or tracheal instillation may act against anti-inflammatory agents, protect against hyperreactivity, delay the appearance of bronchial remodelling and modify the biofilm related with the chronic inflammation caused by certain potentially pathogenic microorganisms. Thus, the treatment with HA in its different formulations might play a key role in airway diseases with a high inflammatory component, such as chronic rhinosinusitis, asthma, chronic obstructive pulmonary disease (COPD), cystic fibrosis (CF) and the bronchiectasis from other etiological causes [10,11,12,13,14]. HA is also used as a mucolytic during bronchoscopy to extract the mucus that obstructs the airways [15].

## 2. Methods

The methodology used in this narrative review consisted of a careful analysis of the literature regarding the role of HA on the management of chronic airway diseases such as upper respiratory tract infections, COPD, asthma, mucociliary clearance alterations, CF and bronchiectasis. The review included original articles, both in adult and children populations, as well as in murine models, identified through a computerised search using PubMed. Search terms included in title/abstract: hyaluronic acid, hypertonic saline, emphysema, asthma, bronchiectasis, elastase, COPD, CF, mucociliary clearance, nasal polyposis, rhinosinusitis and upper respiratory tract infections. The reference sections of the identified articles were also evaluated for further relevant publications, selecting—as done with the initial publications identified—those that better matched with the aims of the present review and were of clinical interest and benefit for the reader.

## 3. Hyaluronic Acid and Respiratory System

### 3.1. Upper Respiratory Tract

The mucociliary system is one of the first defence lines of the airways. Patients with rhinitis or rhinosinusitis have an alteration of the mucus clearance. Different investigations that use in vitro models of airway mucus transport and the epithelial barrier suggest that low-molecular weight HA protects the airway epithelium against the damage caused by bacterial infections by hydrating its surface. Zahn et al. observed, in regards to immunofluorescence and Western blot, a significant dose-dependent increase by low-molecular weight HA (40 kDalton) in the expression of tight junction proteins, as well as an increase in the transepithelial resistance. Additionally, the incubation of airway epithelial cells with hyaluronan 40 kDalton significantly increased the gap junction functionality that protects the airway epithelium against injury induced by bacterial products during infection [16]. Many studies have also demonstrated the safety and efficacy of intranasal HA—aerosolised or by instillation—dissolved in isotonic solution (IS) or hypertonic solution (HS) in patients with rhino-sinusal pathologies that underwent or not sinus surgery [10,17,18].

Topic HA is an interesting treatment formulation that delivers the drug in the nose, increasing the availability of the drug in the pathology site, which enables high local concentrations with minimal adverse events. In Table 1, a list of studies, performed in patients with upper respiratory tract infections (i.e., rhinosinusitis, polyposis, etc.) or to improve nasal turbinate surgery, is included.

Macchi et al. performed a randomised, double-blind, parallel group and placebo-controlled study to evaluate the efficacy of nebulised HA nasal washes administered twice-daily for 15 consecutive days per month, for three consecutive months, in 75 children (>four years of age) with recurrent upper respiratory tract infections. Patients were randomised to receive intranasal 9 mg of HA dissolved in 3 mL of IS or 6 mL of IS alone. The ciliary mortality significantly improved in the HA treatment group vs. the IS alone group. Likewise, the adenoid hypertrophy, rhinitis and nasal dyspnoea also improved, and the presence of bacteria, neutrophils and biofilm creations decreased. Additionally, the HA treatment group showed significantly lower days of absence from school vs. controls [17].

The same authors performed a different study with HA in the post-surgery management of patients that underwent functional endoscopic sinus surgery (FESS). The study—randomised, double-blind, parallel-group and placebo-controlled—included 46 patients (>four years of age) who underwent FESS for rhino-sinusal tract infections. Patients were randomised to receive intranasal 9 mg of HA dissolved in 3 mL of IS or 6 mL of IS alone twice-daily for 15 consecutive days per month for three consecutive months. At the end of the treatment, HA + IS following FESS was associated with significant improvements in nasal dyspnoea, the appearance of nasal mucosa at endoscopy and ciliary mortality compared to IS alone, while improving the presence of post-surgery biofilms. The authors concluded that nebulised HA favours the tissue repair of the nasal mucosa after surgery [19].

The safety and efficacy of intranasal HA in the mucociliary clearance after FESS to treat grade II nasal polyposis was also evaluated. The authors performed a randomised, simple-blind, controlled study involving 36 patients. After surgery, patients were randomised to receive nebulised intranasal 9 mg of HA plus 3 mL or 5 mL of IS alone twice-daily for 30 days. Hyaluronic acid was well-tolerated and improved the mucociliary clearance, with a lower incidence of rhinorrhoea, less nasal obstruction and lower incidence of nasal exudates [10].

Casale et al. evaluated the efficacy of HA adjuvant treatment to improve nasal breath and minimise the discomfort after inferior nasal turbinate surgery. The authors performed a study—prospective, randomised and controlled—involving 57 adult patients that were randomised to receive intranasal 3 mL of HA dissolved in 2 mL of IS or IS alone twice-daily for 15 days from the first postoperative day. The authors concluded that HA improved the nasal breath and minimised the post-surgery discomfort after inferior turbinate hypertrophy, favouring a faster healing of the mucosa than IS alone [20].

In the same line of evaluating the efficacy of HA after surgery, Cantone et al. conducted a randomised, double-blind, controlled study in 122 patients that underwent FESS for medically resistant chronic rhinosinusitis and nasal polyposis. One of the arms was treated with intranasal 9 mg of high-molecular weight HA (3 mL) plus 2 mL of IS and the other arm with 5 mL of IS alone. The treatment was administered twice-daily for 30 days for three months. In conclusion, the authors highlighted that the quality of life (QoL) of HA-treated patients significantly improved in comparison to those treated with IS alone [21].

The tolerability of a hypertonic saline solution (HS) with HA was recently evaluated in patients with chronic sinusitis. The study (single arm, not controlled) enrolled 80 adult patients (>18 years) with chronic sinusitis, instilling two puffs of solution in each nostril twice-daily for 20 days. HS with HA was effective and safe and improved the symptoms of chronic rhinosinusitis, and most patients were rather satisfied with the treatment [18].

### 3.2. Chronic Obstructive Pulmonary Disease

As a result of an alteration in the extracellular matrix, a remodelling of the airways is produced in COPD, affecting their thickness, resistance and elasticity and causing a progressive and irreversible obstruction of the air flow [22].

In vitro [23] and in vivo [24] studies in animals have shown that tobacco exposure reduces the content of HA in the lungs. These investigations were later confirmed in COPD patients [25,26] and in post-mortem studies in patients with alpha-1 antitrypsin (AAT) deficiencies [27]. A potential effect of lysozyme has been also postulated in the genesis of emphysema [28], hindering the ability of low-molecular weight HA to prevent the damage of elastase to the elastic fibres [29]. Thus, a significant increase of elastolysis was observed in the matrix samples treated sequentially with lysozyme and HA, and later cultivated with pancreatic elastase, vs. those treated with HA alone [29]. Laboratory studies indicate that HA binds to elastic fibres, protecting them from elastolysis. The mechanism responsible for the interaction between HA and elastic fibres possibly involved the formation of electrostatic or hydrogen bonds between them. The self-aggregating nature of HA also suggests that both exogenous-administered and native HA may combine to form large molecular complexes.

Papakonstantinou et al. quantified the levels of HA, HA synthetase and hyaluronidase in the bronchoalveolar lavage (BAL) of stable COPD patients (53 patients) and in patients during an exacerbation (44 patients) and compared these levels with those from healthy subjects. HA, HA synthetase and hyaluronidase levels were significantly higher in COPD patients, both in the stable phase and during exacerbations, vs. the levels in healthy subjects, which produced an increase in the low-molecular weight HA fragments. A reverse and significative correlation between the levels of HA and hyaluronidase in the forced expiratory volume in one second (FEV_1_) (% of predicted) was observed in the exacerbations, which indicates that the increase of the degradation of HA can be related with the obstruction grade [30].

Recently, those same authors analysed the HA and its degradation products in serum in two cohorts of COPD stable patients during the exacerbations and four weeks after them. Additionally, they determined the hyaluronidase-1 enzyme and its activity. The authors observed that the HA serum levels in the stable phase were associated with the severity of COPD and predicted the survival of patients, regardless of other known prognostic factors, such as the annual exacerbations rate and Charlson score. The levels were significantly higher during moderate and severe exacerbations than in the clinical stability phase and continued high four weeks after the acute phase was finished. On the other hand, hyaluronidase-1 serum levels were increased during moderate and severe exacerbations but diminished four weeks after the acute phase. Its activity, in the clinical stability phase (increasing the enzymatic degradation of high-molecular weight HA), showed a reverse correlation with FEV_1_ (% of predicted) and mortality [31]. Coinciding with these data, previous research has found short fragments of HA in the lung parenchyma [32] and BAL [32,33] of different respiratory pathologies. These results indicate that HA, as collagens, is a part of the increased extracellular matrix turnover in COPD, a process that determines the disease severity.

Due to research that supported the idea that the depletion of HA in the lungs with emphysema could be a factor that increased the progression of elastolysis and lung emphysema, some studies were designed to evaluate if the exogenous administration of HA could have any protective impact against the elastase activity by forming a barrier that protects the elastic fibres of the lungs. Exogenous HA administration can be combined with the endogen, forming bigger molecular complexes [34]. A related question, for clinical applications, is the ideal size of HA as a treatment agent. A common, and reasonable, concern about the application of high-molecular weight HA in inflammatory disease is that it will be degraded to short-fragment HA, thus “adding fuel to the fire” in the intermediate or long term. Yet, there is no evidence of this effect in either animal models or human studies that employed high-molecular weight HA over several weeks. Additionally, although short-fragment HA does not protect from exercise-induced hyper-responsiveness in human asthma, and induces inflammation in naive mice, it seems to protect from the development of COPD in animal models and is being currently tested in clinical COPD trials. Thus, it appears that we still do not fully understand the scope of HA signalling or its effects in disease. It may be that the pharmacological application of HA through the airway reaches a different compartment than short-fragment HA released in the interstitial space in inflammation, and this may account for the observed differences. Alternatively, it could be that short-fragment HA has adverse effects in naive tissues but acts as an antagonist to stronger inflammatory triggers, such as endotoxin and cigarette smoke. However, these hypotheses are yet to be experimentally tested

In this context, different studies carried out in animal and human models have shown a protective effect of aerosolised HA and its degradation products in the airways remodelling of COPD patients, probably by adherence to the elastic fibres and protecting them against chemical hydrolysis of elastin by the pancreatic and neutrophilic elastases [35,36,37,38,39,40], not chemically inhibiting them [41]. This effect has been demonstrated in vivo in lung emphysema models in hamsters [37], as in vitro in the elastin matrix produced in pleural mesothelial cell cultures [42].

A recent study was performed in 11 patients with COPD to evaluate the safety of aerosolised HA and if its administration modified the degradation of elastin by measuring isodesmosine (Table 2). Eight patients received 0.01% of 150-kDalton HA dissolved in 3 mL of IS twice-daily, and three patients received 3 mL of IS alone. The authors did not find any relevant adverse events and showed that the isodesmosine levels were reduced significantly in patients receiving HA [11].

### 3.3. Asthma

Asthma is characterised, among others, as a chronic inflammation and remodelling of the airways [43]. Recent studies have revealed the importance of HA in the pathogenesis of asthma, proving an increase of the short fragments that produce inflammation and bronchial hyper-reactivity [44,45,46]. Extensive research in the last 20 years has shown that short-fragment HA signalling through receptors CD44 and RHAMM contributes to the accumulation of immune cells in inflammatory sites; furthermore, short-fragment HA activates immune cells and leads to the release of proinflammatory cytokines and metalloelastases and the inhibition of plasminogen activation. Importantly, HA also mediates experimentally induced airway hyper-responsiveness, with a clear size-dependent response. Short-fragment HA, but not high-molecular weight HA or oligosaccharides of HA, replicates the inflammatory changes and hyper-responsiveness in the airway. High-molecular weight HA activates regulatory T cells and promotes the expression of anti-inflammatory cytokines. Instilled high-molecular weight HA ameliorates allergic airway inflammation and bronchial hyper-reactivity with decreasing the formation of the pathological HA matrix and reducing the activation of Rho-associated, coiled-coil containing protein kinase 2 (ROCK2), a kinase that mediates airway hyper-responsiveness in allergic airway inflammation. Additionally, published studies have shown that the inhibition of low-molecular weight HA signalling predominantly affected the eosinophil and macrophage influx with minor effects in the lymphocytes.

Different authors have explored the idea that the inhalation of high-molecular weight HA could reduce bronchial hyper-reactivity. Scuri et al. proved more than a decade ago that aerosolised HA in a dose-dependent and molecular weight-dependent fashion significantly reduced the bronchoconstriction secondary to the inhalation of pancreatic elastase in animals [47,48].

Very few studies are available evaluating the effects of HA in asthma (Table 2). Petrigni et al. in a randomised, single-blind study administering HA or IS (as the placebo) in two nonconsecutive days, 30 min prior to exercise, to 14 patients with mild asthma concluded that the administration of aerosolised HA with a molecular weight variable from 400 to 4000 kDalton significantly reduced the bronchial hyper-reactivity secondary to exercise in asthmatic patients [49]. This protective effect has not been validated by other authors [50]. High-molecular weight HA may have multiple targets against airway inflammation: immune cells and epithelial cells, as has been shown in fibrotic lung injury, and airway myocytes.

In mice models with an allergic inflammation or airways induced by dust mites, Johnson et al. showed that the instillation of a commercially available high-molecular weight HA preparation (Yabro, IBSA International, 6915 Pambio-Noranco, Lugano, Switzerland) can diminish the allergic inflammation and bronchial hyper-reactivity, also administered after the implementation of an allergic sensibilisation. Thus, the authors concluded that the administration of high-molecular weight HA may be a potential treatment for this inflammation [12].

### 3.4. Mucociliary Clearance Alterations

Mucociliary clearance is one of the main defence mechanisms of the respiratory system. Its function is related with a significant number of factors, such as cough, mucus viscosity and the volume of the periciliary liquid. In patients with bronchiectasis secondary or not to CF, the alteration of the mucociliary clearance contributes to the vicious cycle of obstruction-infection-inflammation key in the pathogeny of these diseases. Thus, one of the key treatment objectives of patients with chronic bronchial hypersecretion is to expel the mucus that obstructs the airways.

In CF, HS acts, improving the mucociliary clearance in different ways, such as diminishing the viscosity of the mucus [51], stimulating a cough [52], increasing the hydration of the surfaces of the epithelial cells [53] or inhibiting the epithelial sodium channels (ENaC) [54]. It is not clear if HS per se has anti-inflammatory properties [55,56] but, added or not to respiratory physiotherapy, has demonstrated efficacy and safety in expelling mucus from the airways [57,58]. Furthermore, nebulised twice-daily, at 6% or 7% concentration, it hydrates the airways; diminishes the frequency of exacerbations and improves the lung function, the mucociliary clearance and the quality of life (QoL) of CF patients [59]. As a result of that, CF guidelines recommended HS as a maintenance treatment [60].

The mode of action of HS in bronchiectasis is not clear even when it is considered that, through hydrating the secretions, it favours mucociliary clearance [61]. The level evidence of its efficacy and safety in bronchiectasis or COPD is clearly lower than in CF [62,63,64,65]. Thus, most recommendations related with the use of these solutions in bronchiectasis come from the extrapolation of its efficacy and safety in CF [66,67,68].

Nevertheless, up to 10% of CF patients do not tolerate HS [69], having dry cough, bronchospasm, dyspnoea, throat irritation, chest tightness and the salty flavour of the solution being what challenges the adherence to treatment.

On the other hand, as HS has demonstrated its efficacy, during the last years, certain initiatives have tried to improve the tolerability while maintaining a high salinity (6% to 7%), as lower salt concentrations have not shown a comparable efficacy [69]. In this respect, some studies have been focused on the role that adding HA could have in the tolerability of HS.

### 3.5. Cystic Fibrosis

Since HA appears to prevent bronchoconstriction and protect against inflammatory mediator-induced bronchoconstriction [48,49], several studies with patients with CF have been carried out with the objective of improving the tolerability and decreasing the bronchial hyper-reactivity (Table 3). Additionally, it diminishes the salty flavour of the solution [13,70,71,72].

The first study performed to evaluate the tolerance of HA inhalation added to HS was led by Buonpensiero et al. The trial-open, randomised, crossover included 20 patients with CF and >six years of age and compared one 5-mL dose of 7% HS + HA (0.1% concentration) vs. HS alone in terms of efficacy and safety. The authors concluded that the inhalation of HS + HA was better-tolerated and more pleasant for inhalation than HS alone [13].

Later, Máiz et al. designed a prospective, observational study to evaluate the tolerance to two different HS regimes in CF patients. The study included 81 CF patients with >six that inhaled a 5-mL dose of HS 7% concentration, and those who did not tolerate that dose were tested for tolerance to HS + 0.1% HA (Hyaneb, Eupharma s.r.l., 40121, Bologna, Italy) (5 mL) at least 24 h later. The tolerability was assessed in terms of cough, throat irritation, nausea and dyspnoea, as perceived by the patients within the first hour after the inhalation of the solution. Twenty-one patients (26%) did not tolerate HS, cough being the most common cause for intolerance, followed by throat irritation, nausea and dyspnoea. Seventeen out of the 21 patients (81%) that did not tolerate HS tolerated well HS + HA. All these patients referred to an improvement in the salty flavour by nebulising HS + HA. The authors concluded that HA added to HS mitigated the adverse events of HS alone [71].

In the same working field, Furnari et al. developed a prospective, randomised, double-blind, parallel-group, controlled study involving 20 patients with CF > 10 years. The trial compared the tolerability of 7% HS + 0.1% HA (Hyaneb) (5 mL) or 7% HS (5 mL). In contrast with previous investigations—where the tolerability was evaluated with one dose in this study—after the baseline, patients that tolerated the assigned treatment were treated with this twice a day for 28 days, assessing their tolerability across the entire treatment. As found in previous studies, HS + HA reduced the need of rescue bronchodilators and produced fewer adverse events than HS alone, and the majority of patients expressed a preference for HA [70].

In the last study published with HS + HA, Ros et al. developed a prospective, randomised, double-blind, controlled study involving 40 patients with CF > eight years. The objective of the investigation was to assess whether 7% HS + 0.1% HA was better-tolerated than 7% HS alone in those patients that previously had an even tolerance to HS. Patients were randomly assigned to receive 7% HS, 5 mL or HS + HA, 5 mL twice a day for 28 days. The authors concluded that adding HA to HS reduced the prevalence and severity of the cough, throat irritation and salty flavour in patients that previously showed poor tolerance to HS alone [72].

### 3.6. Bronchiectasis

The available evidence of the effect of HS alone in bronchiectasis is smaller than in CF. The three randomised clinical trials published have remarkable differences in the design, inclusion and exclusion criteria; HS dosing and nebulisers used, which makes it difficult to establish an indirect comparison among them [73]. Adverse events found with HS were very similar to those found in CF patients [62,63,64], even when the percentage of patients that did not tolerate the treatment was higher [14].

As in other pathologies, adding HA to HS without modifying the salinity improved the tolerability to HS. Similar to what happens with CF patients, the precise cause of this reduction in HS adverse events in patients with bronchiectasis is not well-known, although could be related with its hydration properties of the airways and its capacity to diminish the bronchial hyperreactivity and reduce inflammation [74,75]. Due to this capability to mitigate HS adverse events, some treatment guidelines recommend the administration of HA + HS in patients intolerant to HS [66]. In this regard, two clinical trials with clearly different designs evaluating the tolerability of HS + HA in patients with bronchiectasis [14,76] have been recently published, supporting the recommendations included in some guidelines (Table 3).

Herrero-Cortina et al. conducted a single-site, randomised, double-blind, crossed study with three consecutive treatment arms. Each arm was made of four sessions, once a day, separated by a wash-out period of seven days. All sessions except the third included 30 min of respiratory physiotherapy. The study compared 5 mL of three saline solutions (7% HS, 7% HS + 0.1% HA-Hyaneb and IS) in 28 patients with bronchiectasis and chronic expectoration. The most severe adverse events were reported in the HS treatment arm, followed by HS + HA and IS, cough and throat irritation being the most common adverse events reported. The sputum collected in the 24-h follow-up after the sessions was less in the HS and HS + HA groups than in the IS group, suggesting that HS solutions were able to obtain a higher amount of expectorated sputum during the combined sessions. The authors concluded that both HS and HS + HA were more effective than IS in facilitating the expectoration and with a higher safety profile [76].

Máiz et al. performed a multicentric, prospective, open, observational study in 137 patients with bronchiectasis and chronic expectoration to evaluate the tolerance to 7% HS + 01% HA (Hyaneb) (5 mL) in patients intolerant to 7% HS. The tolerance to HS was evaluated by a Likert-like questionnaire self-administered to patients, evaluating the severity of the adverse events by spirometry and by a clinical assessment from the healthcare professionals performing the test. Ninety-two (67.1%) out of the 137 patients initially tolerated HS, but of these, eight (8.7%) did not complete the four-weeks treatment due to a progressive intolerance to the treatment. The most common adverse events were cough, throat irritation and salty flavour. Forty-five patients (32.9%) were intolerant to HS, and 31 of them (68.9%) tolerated HS + HA at the first visit. The baseline, postbronchodilator and post-HS FEV_1_ values were significantly lower in patients who did not tolerate HS than in those who did. No significant differences were found in the baseline, postbronchodilator and post-HS + HA FEV_1_ values between patients who tolerated, and those who did not tolerate, HS + HA. The HS + HA inhalation resulted in less significant adverse events leading a withdrawal of the study than HS alone. The authors concluded that HS + HA improved the tolerance to HS [14].

## 4. Conclusions

Hyaluronic acid is a key component of the extracellular matrix of the lungs and plays a key role in water regulation in the interstitium.

Several studies have highlighted the role that HA and its degradation products play in the physiopathology of the respiratory system. HA administered by aerosol or tracheal instillation acts against inflammation, protecting the airways against the hyper-reactivity and remodelling of the bronchial parenchyma.

It has been determined that the exogen administration of HA can play a significant role in certain airway diseases, such as chronic sinusitis, asthma, COPD, CF and bronchiectasis. In this regard, aerosolised HA has shown its benefit in: the post-surgery management of patients that underwent FESS for chronic rhinitis and nasal polyposis, in children with recurrent infections of the upper respiratory tract, in the remodelling of the airways of COPD patients and in the attenuation of bronchial hyper-reactivity in patients with asthma. In CF and bronchiectasis, HA improves the tolerance to HS attenuating bronchial hyper-reactivity and mitigating the salty flavour of the HS solution.

## Figures and Tables

**Figure 1 cells-09-02210-f001:**
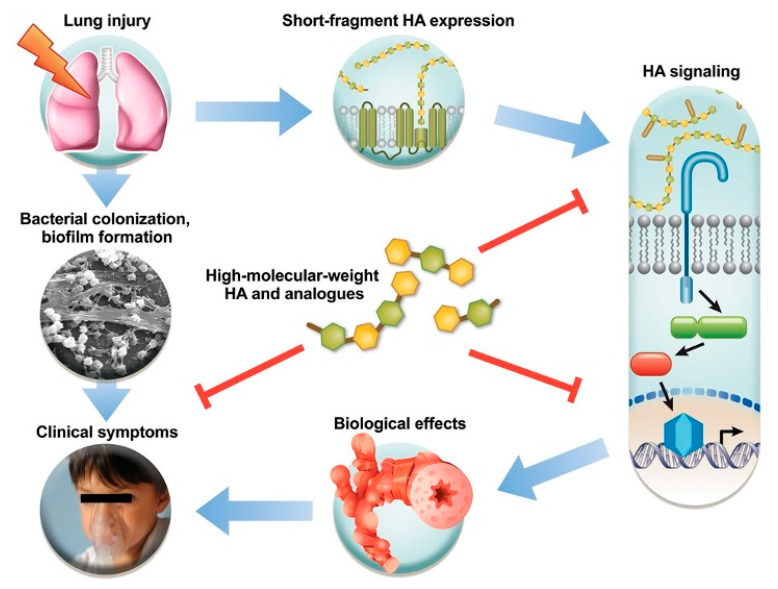
Effects of short-fragment hyaluronic acid (HA) on a lung injury (from Garantziotis S et al. The role of hyaluronan in the pathobiology and treatment of respiratory disease. Am J Physiol Lung Cell Mol Psysiol 2016; 310:L785–L795).

**Table 1 cells-09-02210-t001:** Studies of clinical use of topic HA in upper respiratory tract including respiratory infections, rhinosinusitis, and nasal polyposis.

First Author (Year)	Study Design	N; Male (%); Mean Age	Inclusion and Exclusion Criteria	Intervention	Results	Adverse Events
Macchi (2013)	Prospective, randomised, double-blind, parallel-group, placebo-controlled	75; 40 (53.3%); 13	Inclusion: recurrent upper-respiratory tract infections. Exclusion: history of immunological or autoimmune diseases, genetic syndromes, malformations or presenting with any indication for surgery.	Intranasal 9 mg of HA in 3 mL of IS or 6 mL of IS alone twice-daily for 15 consecutive days per month for 3 consecutive months.	HA nasal washes + IS was superior to IS alone in adenoid hypertrophy, presence of bacteria, neutrophils, rhinitis, nasal dyspnoea and biofilm. The number of days of absence from school was significantly lower in the HA group compared to controls.	Not reported.
Macchi (2013)	Prospective, randomised, double-blind, parallel-group, placebo-controlled	46; 23 (50%); 38.5	Inclusion: patients > 4 years who underwent FESS for rhino-sinusal tract infections. Exclusion: no exclusion criteria.	Intranasal 9 mg of HA in 3 mL of IS or 6 mL of IS alone twice-daily for 15 consecutive days per month for 3 consecutive months.	HA + IS following FESS was associated with significant improvements in nasal dyspnoea, appearance of nasal mucosa at endoscopy and ciliary motility vs. IS alone, while improving the presence of post-surgery biofilms.	HA was well-tolerated.
Gelardi (2013)	Prospective, randomised, simple-blind, controlled	36; 16 (44.4%); 47	Inclusion: patients undergoing FESS to treat grade II nasal polyposis. Exclusion: patients affected by grade I nasal polyposis, CF, primitive ciliary dyskinesia and choanal atresia.	Intranasal 9 mg of HA plus 3 mL or 5 mL of IS alone twice-daily for 30 days on the second day after surgery.	At 1 month, patients receiving HA had a significantly faster mucociliary clearance time, experienced a lower incidence of rhinorrhoea, less nasal obstruction and a lower incidence of exudate on endoscopy than control subjects.	HA was well-tolerated in patients following FESS.
Casale (2013)	Prospective, randomised, controlled	57; (50 completed the study); 25 (43.9%); 45.6	Inclusion: patients with nasal obstruction resulting from inferior turbinate hypertrophy refractory to medical therapy who underwent radiofrequency volumetric tissue reduction. Exclusion: patients with previous turbinate surgery, significant septal deformity, septal perforation, ala collapse, nasal polyposis, benign or malignant tumours of the nasal cavity, nasal radiotherapy and other comorbidities.	Intranasal 3 mL of HA dissolved in 2 mL of IS or IS alone twice-daily for 15 days from the 1st postoperative day.	The mean VAS of the HA group at the 1st week was lower than the IS group and remained significantly lower in the HA group also at the 2nd week. The HA group showed lower endoscopic nasal scores than the IS group, especially for crusts. Compliance in the HA group was lower than in the IS group (76% vs. 98%).	No adverse outcomes related to HA were recorded.
Cantone (2014)	Prospective, randomised, double-blind, controlled	124; (122 completed the study); 70 (56.4%); 41.2	Inclusion: patients undergoing FESS for medically resistant chronic rhinosinusitis and nasal polyposis. Exclusion: patients suffering from systemic diseases, acetylsalicylic acid sensitivity, CF, primitive ciliary dyskinesia, grade I nasal polyposis, choanal atresia and with a history of previous interventions.	Intranasal 9 mg of HA (3 mL) plus 2 mL of IS or 5 mL of IS alone twice-daily for 30 days from the 1st postoperative day.	After postoperative treatment, the endoscopic score, the total VAS score, the mean SNOT-22 and SF-36 results were better in the HS group than in the IS group.	HA was significantly better-tolerated than IS in patients undergoing FESS.
Monzani (2020)	Prospective, single arm, not controlled	87; (86 completed the study); 41 (47.1%); 56	Inclusion: history of previously diagnosed or recurrent chronic rhinosinusitis or a clinical diagnosis of chronic rhinosinusitis. Exclusion: patients who underwent previous sinus-nasal surgery in the last 12 m before study inclusion, with a history or presence of benign and malignant tumours of the nasal cavity, history of recurrent epistaxis, coagulation disorders, evidence or history of chronic bacterial rhinosinusitis and other comorbidities.	Patients were instructed to use HA solution twice-daily for 20 days.	HA was significantly effective in the relief of symptoms of recurrent chronic rhinosinusitis. Nasal blockage, nasal congestion, nasal drainage and rhinorrhoea improved. Middle turbinate oedema and nasal secretion at nasal endoscopic evaluation significantly improved.	No secondary effects related with HA were reported.

HS, hypertonic saline, IS, isotonic saline, HA, hyaluronic acid, CF, cystic fibrosis, FESS, functional endoscopic sinus surgery, VAS, visual analogue scale, SNOT-22, Sino-nasal outcome test-22 and SF-36, short form-36.

**Table 2 cells-09-02210-t002:** Studies of the clinical use of nebulised hyaluronic acid in chronic obstructive pulmonary disease and asthma in humans.

First Author (Year)	Study Design	N; Male (%); Mean Age	Inclusion and Exclusion Criteria	Intervention	Results	Adverse Events
**COPD**						
Cantor (2017)	Prospective, randomised, double-blind, placebo-controlled	11; no data; 60	Inclusion: COPD patients with GOLD grades 2 and 3 with moderate airway obstructions and at least a 10-pack/year history of cigarette smoking. Exclusion: no active smoking.	Each patient self-administered 3 mL of aerosolised inhalation solution (0.01% of HA in 3 mL of IS or 3 mL of IS alone) twice-daily for 14 days.	Measurements of desmosine and DID in plasma from HA-treated patients indicated a progressive decrease over a 3-week period following initiation of the treatment. There was no significant reduction in the placebo group. Patients receiving IS showed no reduction in DID. Measurements of sputum in the HA-treated group revealed a progressive decrease in DID.	HA was well-tolerated and did not involve adverse events requiring the cessation of treatment.
**ASTHMA**						
Petrigni (2006)	Prospective, randomised, cross-over, single-blind	14; 11 (78.6%); 21.36	Inclusion: patients with mild persistent bronchial asthma. Exclusion: no active smoking, patients participated in the study out of the season of their individual allergy.	A single dose of IS as placebo (4 mL) or HA (iso-osmolar solution containing 0.3% of HA) was administered by aerosol in 2 nonconsecutive days, 30 min prior to the beginning of the challenge (10 min free running).	Pretreatment induced with aerosolised HA determined partial but clear protection on the FEV_1_ reduction due to the bronchoconstriction exercise.	No data.
Kunz (2005)	Prospective, randomised double-blinded placebo-controlled crossover	16; 6 (37.5%); no data	Inclusion: clinical history of asthma, clinically stable lung disease 2 weeks prior to the screening, FEV_1_ ≥ 50% predicted value, concentration methacholine at which the patient had a fall in FEV_1_ of 20% of <8 mg/mL and > 15% fall from baseline FEV_1_ within 30 min after an exercise challenge. Exclusion: nonsmoking.	On 2 separate visits, an exercise challenge was performed 15 min post-inhalation of either HA (3 mL 0.1% in PBS) or placebo (3-mL PBS). The wash-out period between both treatment days was 7–14 days.	The maximum fall in FEV_1_ following the exercise challenge was not significantly different between HA vs. placebo, as was the area under the time-response curve.	HA was well-tolerated, and no serious adverse events were reported.

COPD, chronic obstructive pulmonary disease; GOLD, global initiative for chronic obstructive lung disease; HA, hyaluronic acid; IS, isotonic saline; DID, isodesmosine; FEV_1_, forced expiratory volume in one second and PBS, phosphate-buffered saline.

**Table 3 cells-09-02210-t003:** Clinical studies of use of hyaluronic acid plus hypertonic solution in cystic fibrosis and bronchiectasis.

First Author (Year)	Study Design	N; Male (%); Mean Age	Inclusion and Exclusion Criteria	Intervention	Results	Adverse Events
**Cystic Fibrosis**						
Buonpensiero (2010)	Prospective, open, randomised, crossover trial, one daily session	20; 9 (45%); 13	Inclusion: patients with CF, ≥6 years, FEV_1_ ≥ 50% predicted value, clinically stable lung disease. Exclusion: evidence of reactive airways or a clinical diagnosis of asthma.	One dose of 7% HS, 5 mL or HS + HA, 5 mL.	↓ cough, throat irritation and salty taste in HS + HA group than in IS group. More pleasant ratings of taste in HS + HA group than in IS group.	HS + HA inhalation produced less significant adverse events than the HS group.
Máiz (2012)	Prospective, observational	81; 44 (54.3%); 23.6	Inclusion: patients with CF, >6 years, clinically stable. Exclusion: an exacerbation in the 15 days preceding inclusion, patients unable to perform a spirometry test and those with a history of haemoptysis due to the use of nebulised drugs.	Tolerance to HS (5 mL) was first assessed. Patients nontolerant to one dose of HS were tested for tolerance to HS + HA (5 mL) at least 24 h later.	Twenty-one (26%) patients did not tolerate the HS solution immediately after its inhalation. Eighty-one percent of patients who did not tolerate the HS alone tolerated well the HS + HA. Patients ≥ 18 years of age showed the worst tolerance to HS than patients younger than 18 years. Those patients that did not tolerate HS had worse lung functions than the ones that showed good tolerance.	HS + HA inhalation produced less significant AEs than the HS group.
Furnari (2012)	Prospective, randomised, double-blind, parallel group, controlled	30 (27 completed the study); 16 (53.3%); 23.2	Inclusion: patients with CF > 10 years, FEV_1_ ≥ 40% predicted vale, clinically stable, in the 3 months prior to study inclusion. Exclusion: *Burkholderia cepacia* infection, used HS therapy in the 15 days preceding enrolment.	7% HS, 5 mL or HS + HA, 5 mL twice a day, 28 days.	HS + HA was more effective in reducing cough, throat irritation and incidence of bronchoconstriction. The overall judgment of treatment pleasantness was significantly different in favour of the HS + HA group compared with the HS group. The consumption of bronchodilators was statistically significantly lower in the HS + HA group compared with the HS group.	No AEs were reported in either group during the study.
Ros (2013)	Prospective, randomised, double-blind, parallel group	40 (35 complete the study); 16 (40%); 24	Inclusion: patients with CF ≥ 8 years, clinically stable disease during the previous 30 days, FEV_1_ ≥ 50% predicted value, intolerance to HS solution. Exclusion: decrease in FEV_1_ of >15% after HS, infection with *Burkholderia cepacian*, noncompliance to standard therapy, having received lung transplantation, being unable to perform reproducible spirometry, being intolerant to β_2_ bronchodilators, having circulated plasmatic creatinine or transaminase levels.	7% HS, 5 mL or HS + HA, 5 mL twice a day, 28 days.	Severity of cough, throat irritation and saltiness were more severe in patients treated with HS alone.	The prevalence and severity of the secondary effects was higher in the HS group than the HS + HA group.
**Bronchiectasis**						
Herrero-Cortina (2018)	Randomised, double blind, crossed; 3 consecutive treatment branches, 4 daily sessions each, separated by a 7-day washout period	28 (23 completed the study); 10 (35.7%); 64	Inclusion: patients with bronchiectasis diagnosed by HRCT, clinically stable in the last 4 weeks, who spontaneously expectorate ≥ 10 g/day of sputum, able to inhale solutions and perform physiotherapy techniques. Exclusion: active smokers or former smokers, bronchial hyperresponsiveness, ABPA, post bronchodilation FEV_1_ < 30%, TLC < 45% and inhalation of muco-active agents before screening.	3 randomised treatment arms (7% HS, 5 mL, HS + HA, 5 mL and IS, 5 mL), preceded by bronchodilator. All sessions included 30 min of respiratory physiotherapy, except the third.	↑ sputum weight obtained in HS and HS + HA groups than in the IS group. ↓ sputum collected in the 24-h follow-up in HS and HS + HA groups than the IS group. ↑ amount of expectorated sputum during combined sessions than during the sessions with no physiotherapy. No differences in LCQ or lung function were observed.	Most adverse events were in the HS group, followed by the HS + HA and IS groups. Most common adverse event was throat irritation. There were no deaths in any group.
Máiz (2018)	Prospective, observational, open	137; 50 (36.5%); 63	Inclusion: patients > 18 years with bronchiectasis diagnosed by HRCT, sputum production > 30 mL/day, postbronchodilator FEV_1_ <1 L or < 35%. Exclusion: treatment with antibiotics or oral corticosteroids in 4 weeks prior to the study, previous episodes of haemoptysis caused by inhaled drugs, ABPA, CF, patients unable to perform spirometry, pregnant women or uncontrolled high blood pressure.	Tolerance to HS (5 mL) was first assessed. Patients nontolerant to one dose of HS were tested for tolerance to HS + HA (5 mL) a week later. All patients were evaluated for tolerance to treatment one month after the start of treatment.	Sixty-seven point one percent of patients (92) initially tolerated HS. Of these, 8 (8.7%) did not complete the 4-week treatment due to progressive intolerance. Of the 45 patients nontolerant to HS, 31 (68.9%) tolerated HS + HA at the first visit. Of those 31 patients, 1 did not complete the 4-week treatment due to progressive intolerance to the solution. Although both treatments improved the QoL of patients in 7 of the 8 dimensions in the QoL and in the LCQ, no significant differences were found between them for any of the two questionnaires.	HS + HA inhalation produced less significant adverse events than the HS that caused patients to abandon the treatment. There were no deaths in any group.

HRCT, high-resolution computed tomography; ABPA, allergic bronchopulmonary aspergillosis; FEV_1_, forced expiratory volume in one second; TLC, total lung capacity; HS, hypertonic saline; HA, hyaluronic acid; IS, isotonic saline; LCQ: Leicester cough questionnaire; CF, cystic fibrosis; QOL, Quality of Life, Questionnaire Bronchiectasis and AEs, adverse events.

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
