# Peer review of "Use of Hyaluronic Acid (HA) in Chronic Airway Diseases"

_cells, 2020, doi:10.3390/cells9102210_

Round 1

Reviewer 1 Report

Comments to cells-893251

This manuscript describes the effects of Hyaluronic acid (HA) on chronic airway disease.  My impression is the review is rather phenomenological.  I recommend the authors to   add any scientific discussions on the mechanisms of actions of HA.

Major comments

Throughout the text: The actions of high-molecular-weight HA (HHA) and low-molecular-weight HA (LHA) are entirely different.  In Introduction, the authors defined the molecular weights (MWs) of HHA and LHA are >1 million Daltons and 150,000-300,000 Daltons, respectively.  Is this definition applicable to the MW of HHA and LHA throughout the text?  I recommend the authors to describe each MW of HA dealt with each reference if the MW is outside of the definition.

Introduction: This section is very important to describe general effect of HAs on lungs as well as other organs or cells.  I recommend the authors to add more information on the mechanisms of actions of HHA and LHA.

Introduction: Fig. 1 is very informative to describe the effects of HHA and LHA.  However, there is little description about Fig. 1 in the text.  In recommend the authors to describe carefully the effects of HHA and LHA based on Fig. 1.

Particular comments

48-49: What molecular weights of HA do act against anti-inflammatory agents, protect against hyperreactivity, delay the appearance of bronchial remodeling, and modify the biofilm related with the chronic inflammation?

77: The finding that low molecular weight HA protects the airways epithelium against the damage caused by bacterial infections by hydrating its surface is unique compared to the other reports on LHA.  Please add any discussion on this finding.

97: SSI is used without definition.

Table 1: There are some typos. The % of male for the second Macchi (2013) is not 54.3% (44/46).  The % of male for the Cantone (2014) is not 49.3% (70/124). In the footnote, HA should be hyaluronic acid instead of hyaluronic acid solution.

121 and Table 1 (Casale): "3 ml of HA disslved in 2 mL of IS" is not appropriate quantitatively.  What concentration was the HA solution?

145: How and what molecular weight of HA does prevent the damage of elastase to the elastic fibers?

171: Why is the HA turnover essential in the development, progression and resolution of the inflammatory respiratory diseases?

179: It is unique that degradation products as well as aerosolised HA have protective effect in the air ways remodeling of COPD patients.  Any discussion?

197: Why/how do the short fragments produce inflammation and bronchial hyperreactivity?

199: Why/how could inhalation of HHA reduce bronchial hyperreactivity? How about LHA?

210 Why/how can the instillation of HHA diminish the allergic inflammation and bronchial hyperreactivity?

221: HS is used without definition.

240: Why have some studies been focused in the role that adding HA could have in the tolerability of HS? Were there any scientific backgrounds?

275 and Table 3 (Ros): 47 patients in line 275, while 40 in Table 3.

Author Response

Major comments

C1. Throughout the text: The actions of high-molecular-weight HA (HHA) and low-molecular-weight HA (LHA) are entirely different.  In Introduction, the authors defined the molecular weights (MWs) of HHA and LHA are >1 million Daltons and 150,000-300,000 Daltons, respectively.  Is this definition applicable to the MW of HHA and LHA throughout the text?  I recommend the authors to describe each MW of HA dealt with each reference if the MW is outside of the definition.

R1: In the text, now we describe each MW of HA dealt with each reference if the MW is outside of the definition (if data of MW of HA is present in the original article).

For example, in the text (lines 77 to 83): “The mucociliary system is one of the first defense lines of the airways. Patients with rhinitis or rhinosinusitis have an alteration of the mucus clearance and different investigations suggest that low molecular weight HA protects the airways epithelium against the damage caused by bacterial infections by hydrating its surface[16].” has been replaced by (lines 88-91) “The mucociliary system is one of the first defense lines of the airways. Patients with rhinitis or rhinosinusitis have an alteration of the mucus clearance and different investigations suggest that low-molecular-weight HA protects the airways epithelium against the damage caused by bacterial infections by hydrating its surface[16].

Lines 188-191: “A recent study has been performed in 11 patients with COPD to evaluate the safety of aerosolised HA and if its administration modified the degradation of elastin by measuring isodesmosine (Table 2). Eight patients received 0,01% HA dissolved in 3 ml of IS, twice daily, and 3 patients received 3 ml of IS alone” has been replaced by “A recent study has been performed in 11 patients with COPD to evaluate the safety of aerosolised HA and if its administration modified the degradation of elastin by measuring isodesmosine (Table 2). Eight patients received 0,01% of 150 kDalton HA dissolved in 3 ml of IS, twice daily, and 3 patients received 3 ml of IS alone.”

Lines 202-207. “Different authors have explored the idea that the inhalation of high-molecular-weight HA could reduce bronchial hyperreactivity. Scuri et al. proved more than a decade ago that aerosolised high-molecular-weight HA significantly reduced the bronchoconstriction secondary to inhalation of pancreatic elastase in animals[47][48].” has been replaced by “Different authors have explored the idea that the inhalation of high-molecular-weight HA could reduce bronchial hyperreactivity. Scuri et al. proved more than a decade ago that aerosolised HA in a dose-dependent and molecular weight-dependent fashion significantly reduced the bronchoconstriction secondary to inhalation of pancreatic elastase in animals[47][48].”

Lines 207-211: “Very few studies are available evaluating the effect of HA in asthma (Table 2). Petrigni et al. in a randomised, single-blind study administering HA or IS (as placebo) in two non-consecutive days, 30 minutes prior to exercise to 14 patients with mild asthma; concluded that the administration of aerosolised HA significantly reduced the bronchial hyperreactivity secondary to exercise in asthmatic patients[49].” has been replaced by “Very few studies are available evaluating the effect of HA in asthma (Table 2). Petrigni et al. in a randomised, single-blind study administering HA or IS (as placebo) in two non-consecutive days, 30 minutes prior to exercise to 14 patients with mild asthma; concluded that the administration of aerosolised HA with a molecular weight variable from 400 to 4000 kDalton significantly reduced the bronchial hyperreactivity secondary to exercise in asthmatic patients[49].”

Lines 213-215. “In a mice model with an allergic inflammation or airways induced by dust mites, Johnson et al. showed that the instillation of high-molecular-weight HA can diminish the allergic inflammation and bronchial hyperreactivity, also administered after the implementation of an allergic sensibilisation. Thus, authors conclude that the administration of high-molecular-weight HA may be a potential treatment for this inflammation[12].” has been replaced by “In a mice model with an allergic inflammation or airways induced by dust mites, Johnson et al. showed that the instillation of a commercially available high-molecular-weight HA preparation (Yabro, IBSA International, Switzerland) can diminish the allergic inflammation and bronchial hyperreactivity, also administered after the implementation of an allergic sensibilisation. Thus, authors conclude that the administration of high-molecular-weight HA may be a potential treatment for this inflammation[12].”

Lines 263-266. “Later, Máiz et al. designed a prospective, observational study to evaluate the tolerance to two different HS regimes in CF patients. The study included 81 CF patients with > 6 that inhaled a 5 ml dose of HS 7% concentration and those who did not tolerate that dose were tested for tolerance to HS + 0.1% HA (5 ml) at least 24 h later.” has been replaced by “Later, Máiz et al. designed a prospective, observational study to evaluate the tolerance to two different HS regimes in CF patients. The study included 81 CF patients with > 6 that inhaled a 5 ml dose of HS 7% concentration and those who did not tolerate that dose were tested for tolerance to HS + 0.1% HA (Hyaneb, Eupharma s.r.l., Bologna, Italy) (5 ml) at least 24 h later.”

Lines 273-275. “In the same working field, Furnari et al. developed a prospective, randomised, double-blind, parallel-group, controlled study involving 20 patients with CF > 10 years. The trial compared the tolerability of 7% HS + 0.1% HA (5 ml) or 7% HS (5 ml).“ has been replaced by “In the same working field, Furnari et al. developed a prospective, randomised, double-blind, parallel-group, controlled study involving 20 patients with CF > 10 years. The trial compared the tolerability of 7% HS + 0.1% HA (Hyaneb) (5 ml) or 7% HS (5 ml). “

Lines 306-308. “All sessions except the third included 30 minutes of respiratory physiotherapy. The study compared 5 ml of 3 saline solutions (7% HS, 7% HS+0.1% HA and IS) in 28 patients with bronchiectasis and chronic expectoration.” has been replaced by “All sessions except the third included 30 minutes of respiratory physiotherapy. The study compared 5 ml of 3 saline solutions (7% HS, 7% HS+0.1% HA -Hyaneb- and IS) in 28 patients with bronchiectasis and chronic expectoration.”

Lines 315-317. “Máiz et al. performed a multicentric, prospective, open, observational study in 137 patients with bronchiectasis and chronic expectoration to evaluate the tolerance to 7% HS + 01% HA (5 ml) in patients intolerant to 7% HS.” has been replaced by “Máiz et al. performed a multicentric, prospective, open, observational study in 137 patients with bronchiectasis and chronic expectoration to evaluate the tolerance to 7% HS + 01% HA (Hyaneb) (5 ml) in patients intolerant to 7% HS.”

C2. Introduction: This section is very important to describe general effect of HAs on lungs as well as other organs or cells.  I recommend the authors to add more information on the mechanisms of actions of HHA and LHA.

R2. We added in the introduction (lines 50-57) “HA mitigates the action of elastases such as porcine pancreatic elastase, as well as human neutrophil elastase and human macrophage metalloelastase. This has been demonstrated in animal models of pulmonary emphysema, as well as in elastin matrices produced by pleural mesothelial cell culture. Studies have indicated that small fragments of HA contribute to the immune cell response binding to specific cell surface receptors. Low-molecular-weight fragments stimulate mouse alveolar macrophages to produce several cytokines including metalloelastase. High-molecular-weight fragments suppressed such expression”.

However, this topic will be dealt in depth in other articles of this volume.

C3. Introduction: Fig. 1 is very informative to describe the effects of HHA and LHA.  However, there is little description about Fig. 1 in the text.  In recommend the authors to describe carefully the effects of HHA and LHA based on Fig. 1.

R3. Effectively, as the reviewer suggest, now we have added to the text the following paragraph, in the lines 45-50 “Lung damage produces the liberation of short-fragment HA. This release activates innate immune receptors which can result in inflammation, remodeling, and hyperresponsiveness, in addition to other clinical symptoms. The defective clearance in injury results in the development of bacterial biofilms. The perpetuation of this vicious cycle can be altered through the modulation of short-fragment HA signaling, for instance with high-molecular-weight HA”.

Particular comments

C4. 48-49: What molecular weights of HA do act against anti-inflammatory agents, protect against hyperreactivity, delay the appearance of bronchial remodeling, and modify the biofilm related with the chronic inflammation?

R4. “Furthermore, administered via aerosol or tracheal instillation may act against anti-inflammatory agents, protect against hyperreactivity, delay the appearance of bronchial remodeling, and modify the biofilm related with the chronic inflammation caused by certain potentially pathogenic microorganisms.” has been replaced by “Furthermore, high-molecular-weight HA administered via aerosol or tracheal instillation may act against anti-inflammatory agents, protect against hyperreactivity, delay the appearance of bronchial remodeling, and modify the biofilm related with the chronic inflammation caused by certain potentially pathogenic microorganisms” (lines 58-61).

C5. 77: The finding that low molecular weight HA protects the airways epithelium against the damage caused by bacterial infections by hydrating its surface is unique compared to the other reports on LHA.  Please add any discussion on this finding.

R5. “The mucociliary system is one of the first defense lines of the airways. Patients with rhinitis or rhinosinusitis have an alteration of the mucus clearance and different investigations suggest that low molecular weight HA protects the airways epithelium against the damage caused by bacterial infections by hydrating its surface[16].” has been replaced by “The mucociliary system is one of the first defense lines of the airways. Patients with rhinitis or rhinosinusitis have an alteration of the mucus clearance. Different investigations that use in vitro models of airway mucus transport and epithelial barrier suggest that low-molecular-weight HA protects the airways epithelium against the damage caused by bacterial infections by hydrating its surface. Zahn et al. observed that in regard to immunofluorescence and western blot, a significant dose-dependent increase by low molecular weight HA (40 kDalton) in the expression of tight junction proteins, as well as an increase in the trans-epithelial resistance. Additionally, incubation of airway epithelial cells with hyaluronan 40 kDalton significantly increased the gap junction functionality as protects the airway epithelium against injury induced by bacterial products during infection [16]” (lines 88-97).

C6. 97: SSI is used without definition.

R6. Done, thanks.

C7. Table 1: There are some typos. The % of male for the second Macchi (2013) is not 54.3% (44/46).  The % of male for the Cantone (2014) is not 49.3% (70/124). In the footnote, HA should be hyaluronic acid instead of hyaluronic acid solution.

R7. Corrected. Thanks.

C8. 121 and Table 1 (Casale): "3 ml of HA dissolved in 2 mL of IS" is not appropriate quantitatively. What concentration was the HA solution?

R8. Regarding material and methods, authors write: “The HA group (22 patients; 8 women and 14 men; mean age, 45 years; range, 18–72 years), which included patients who received HA (Yabro, IBSA Farmaceutici Italia, Lodi, Italy) 3 mL of HA is dissolved in 2 mL of isotonic solution twice a day through Rinowash (Air Liquide Medical Systems S.p.A., Bovezzo, Italy) (a nebulizer designed to treat upper airway structures, creating micronized particles).” The authors do not provide other data.

C9. 145: How and what molecular weight of HA does prevent the damage of elastase to the elastic fibers?

R9. “A potential effect of lysozyme has been also postulated in the genesis of emphysema[28], hindering the ability of HA to prevent the damage of elastase to the elastic fibers[29]. Thus, a significant increase of elastolysis was observed in the matrix samples treated sequentially with lysozyme and HA, and later cultivated with pancreatic elastase, versus those treated with HA alone[29]” has been replaced by “A potential effect of lysozyme has been also postulated in the genesis of emphysema[28], hindering the ability of low-molecular-weight HA to prevent the damage of elastase to the elastic fibers[29]. Thus, a significant increase of elastolysis was observed in the matrix samples treated sequentially with lysozyme and HA, and later cultivated with pancreatic elastase, versus those treated with HA alone[29]”.

Also, in the text we add the following lines: “Laboratory studies indicate that HA binds to elastic fibers, protecting them from elastolisis. It is possible that the mechanism responsible for the interaction between HA and elastic fibers is the formation of electrostatic or hydrogen bonds that occur between them. The self-aggregating nature of HA also suggest that both exogenous administered and native HA may combine to form a large molecular complexes”.  (lines 163-171).

C10. 171: Why is the HA turnover essential in the development, progression and resolution of the inflammatory respiratory diseases?

R10. We have changed the phrase “Coinciding with these data, previous research has found short fragments of HA in the lung parenchyma[32], and BAL[32][33] of different respiratory pathologies, indicating that the HA turnover is essential in the development, progression and resolution of the inflammatory respiratory diseases.” with: “Coinciding with these data, previous research has found short fragments of HA in the lung parenchyma[32], and BAL[32][33] of different respiratory pathologies. These results indicate that HA, as collagens, is a part of the increased extracelular matrix turnover in COPD, a process that determines disease severity”. (Lines 191-195).

C11. 179: It is unique that degradation products as well as aerosolised HA have protective effect in the airways remodeling of COPD patients.  Any discussion?

R11. We added in the text: “A related question, for clinical applications, is the ideal size of HA as a treatment agent. A common, and reasonable, concern about the application of high-molecular-weight HA in inflammatory disease is that it will be degraded to short-fragment HA, thus “adding fuel to the fire” in the intermediate or long term. Yet there is no evidence of this effect in either animal models or human studies that employed high-molecular-weight HA over several weeks. Additionally, although short-fragment HA does not protect from exercise-induced hyperresponsiveness in human asthma, and induces inflammation in naive mice, it seems to protect from the development of COPD in animal models and is being currently tested in clinical COPD trials. Thus it appears that we still do not fully understand the scope of HA signaling or effects in disease. It may be that pharmacological application of HA through the airway reaches a different compartment than short-fragment HA released in the interstitial space in inflammation, and this may account for the observed differences. Alternatively, it could be that short-fragment HA has adverse effects in naive tissues but acts as an antagonist to stronger inflammatory triggers, such as endotoxin and cigarette smoke. However, these hypotheses are yet to be experimentally tested.” (Lines 201-215).

C12. 197: Why/how do the short fragments produce inflammation and bronchial hyperreactivity?

R12. We added in the text: “Extensive research in the last 20 years has shown that short-fragment HA signalling through receptors CD44 and RHAMM contributes to the accumulation of immune cells in inflammatory sites; furthermore, short-fragment HA activates immune cells and leads to the release of proinflammatory cytokines and metalloelastases and the inhibition of plasminogen activation. Importantly, HA also mediates experimentally induced airway hyperresponsiveness, with a clear size-dependent response. Short-fragment HA, but not high-molecular-weight HA or oligosaccharides of HA, replicates the inflammatory changes and hyperresponsiveness in the airway.” (Lines 236-243).

C13. 199: Why/how could inhalation of HHA reduce bronchial hyperreactivity? How about LHA?

R13. We added in the text: “High-molecular-weight HA activates regulatory T cells and promotes expression of antiinflammatory cytokines. Instilled high-molecular-weight HA ameliorates allergic airway inflammation and bronchial hyperreactivity with a decreasing the formation of the pathological HA matrix and reducing activation of Rho-associated, coiled-coil containing protein kinase 2 (ROCK2), a kinase that mediates airway hyperresponsiveness in allergic airway inflammation. Also, published studies have shown that inhibition of low-molecular-weight HA signaling predominantly affected eosinophil and macrophage influx with minor effects in lymphocytes.” (Lines 243-250).

C14. 210 Why/how can the instillation of HHA diminish the allergic inflammation and bronchial hyperreactivity?

R14  We added in the text: “High-molecular-weight HA may have multiple targets against airway inflammation: immune cells, epithelial cells, as has been shown in fibrotic lung injury; and airway myocytes.” (Lines 262-263).

C15. Line 221: HS is used without definition.

R15. Corrected. Thanks .

C16. 240: Why have some studies been focused in the role that adding HA could have in the tolerability of HS? Were there any scientific backgrounds?

R16. In the text we substitute: “Generally speaking, the addition of HA to HS improve its tolerability by attenuating the bronchial hyper-response[70] and diminishing the salty flavor of the solution[13][71][70][72]. Even when the mechanism trough HA presents these beneficial effects is not known with precision, it is believed that is due to the improvement that induces in the mucus transportation by increasing the volume of the periciliary liquid hydrating the bronchial mucus.” with: “Since HA appears to prevent bronchoconstriction and protect against inflammatory mediator-induced bronchoconstriction [48,49], several studies with patients with CF have been carried out with the objective of improving tolerability and decreasing bronchial hyperreactivity (Table 3). Also, it diminishes the salty flavour of the solution[13][71][70][72]”. (lines 299-302).

C17. 275 and Table 3 (Ros): 47 patients in line 275, while 40 in Table 3.

R17. The reviewer is right. It was a mistake and now has been corrected. Thanks.

Reviewer 2 Report

This is the paper to review the role of HA treatment in the management of chronic airway diseases
and authors extensively reviewed the role of HA treatment in the upper respiratory tract disease, asthma, COPD, BE and CF.

The use of HA in hronic airway diseases is not well-know thus, it is worth for readers to know the role of HA in chronic disease.

The authors made the extensive reviews for the role of HA in the lung diseases.
This paper provided the therapeutic information of unknown property of HA in lung diseases.

Additionally, (I misunderstood the meaning of stars)
Is the work a significant contribution to the field? 4 stars
Is the work well organized and comprehensively described? 4 stars
Is the work scientifically sound and not misleading? 4 stars
Are there appropriate and adequate references to related and previous work? 4 stars
Is the English used correct and readable? 4 stars

Minor comments

1. please spell out the full term when you use an abbreviation such as SSI (line 97)
2. At Table 1, clarify the nmber of N;male(%) mean age with same decimal places.

Author Response

Reviewer 2

1) Please spell out full term when you use an abbreviation such as SSI (line 97).

- Replay: done. Thanks.

2) At Table 1, clarify the number of N;mal(%) mean age when the same decimal places.

- Reply: done.

Round 2

Reviewer 1 Report

I agree that the authors corrected the manuscript soundly.